# Nutritional Analysis of the Spanish Population: A New Approach Using Public Data on Consumption

**DOI:** 10.3390/ijerph20021642

**Published:** 2023-01-16

**Authors:** Isabel Cerrillo, Pablo Saralegui-Díez, Rubén Morilla-Romero-de-la-Osa, Manuel González de Molina, Gloria I. Guzmán

**Affiliations:** 1Department of Molecular Biology and Biochemistry Engineering, Area of Nutrition and Food Sciences, Pablo de Olavide University, Carretera de Utrera Km 1, 41013 Seville, Spain; 2Alimentta, Think Tank para la Transición Alimentaria, 18320 Santa Fe, Spain; 3Laboratory of the History of Agroecosystems, Pablo de Olavide University, Carretera de Utrera Km 1, 41013 Seville, Spain; 4Department of Nursing, Faculty of Nursing, Physiotherapy and Podiatry, Universidad de Sevilla, 41015 Seville, Spain; 5Hospital Universitario Virgen del Rocío, Instituto de Biomedicina de Sevilla, Spanish National Research Council (CSIC), Universidad de Sevilla, 41015 Seville, Spain; 6Centro de Investigación Biomédica en Red de Epidemiología y Salud Pública (CIBERESP), Hospital Universitario Virgen del Rocío, 41015 Seville, Spain

**Keywords:** nutritional status, diet food and nutrition, nutrition policy, public health, public sector

## Abstract

Official population consumption data are frequently used to characterize the diet of countries; however, this information may not always be representative of reality. This study analyses the food consumption of the Spanish population by reconstructing the whole food chain. The results have been compared with the data provided by the National Consumption Panel to which the food losses/waste reported in the literature along the distribution chain have been added. The difference between them allowed a new calculation of the estimated food consumption that was subjected to a dietary-nutritional analysis. Most of the foods were consumed more than those officially reported (range of 5–50%). The unhealthy ratios of consumed foods and recommended servings were: meat products (Rcr = 3.6), fruits and legumes (Rcr = 0.5), and nuts (Rcr = 0.14). Caloric intake surpasses needs. The results were consistent with the data on the prevalence of overweight and obesity in Spain, as well as with the prevalence of associated diseases. To make a judgment about the quality of a country’s diet, it is necessary to have reliable data on food consumption, as well as energy and nutrient intake. This study encourages other authors to implement this method to verify and quantify the possible difference between official and real consumption data.

## 1. Introduction

Over the past 50 years, dietary habits in industrialised countries have been changing and worsening from a nutritional viewpoint [1]. According to the report recently published by the EAT-Lancet Commission, higher intakes of unhealthy foods, such as red meat and sugar, instead of healthy ones, such as nuts, fruits, vegetables and legumes, are contributing to a rising prevalence of obesity and diet-related non-communicable diseases (NCDs), including coronary heart disease, stroke, and diabetes [2]. Spain is a representative case of these worsening dietary patterns. The country currently presents notable rates of overweight and obesity, as well as a high prevalence of chronic diseases associated with a poor quality diet. According to the latest data published by the Organisation for Economic Co-operation and Development (OECD), 53% of adults in Spain are overweight [3]. The child and youth population in Spain also presents worrying data, with 34.1% of the young population aged 5 to 19 years being overweight or obese, and 40.6% of the children aged 6 to 9 years being overweight according to the ALADINO 2019 study [4]. Additionally, the prevalence of chronic diseases associated with overweight or obesity, such as cardiovascular diseases, diabetes type 2, or some cancer diseases [5,6] is very high, entailing a high socioeconomic cost and a greater risk of mortality. The latest data from the Spanish Ministry of Health and Consumer Affairs indicated the following figures: 19% hypercholesterolemia, 17% hypertension, and 7% diabetes in Spain [7]. It has also recently been observed that obese patients suffer worse outcomes after contracting the COVID-19 infection [8]. Previous studies have revealed that food consumption changes in Spain would lead to decreasing the environmental impact of current dietary patterns [9,10] and would also reduce some chronic pathologies such as cardiovascular diseases [11,12].

However, assessments of the Spanish diet based on data from the household consumption surveys of the Spanish Ministry of Agriculture, Fisheries and Food [13,14] might not fully explain the prevalence of overweight or obesity reported. Some studies have also shown differences in the data on actual dietary intake, calculated using FAO food balances or apparent food consumption [15,16,17]. Other studies, based on direct consumer surveys, also show contradictory results [18,19,20]. On this basis, our current hypothesis is that there exists a possible inaccurate estimation of consumption data, which results in an overall approach to Spanish dietary-nutritional health missing to justify the current disease prevalence, as databases do not collect real tendencies and patterns. Our approach wants to fill this knowledge gap as it tries to identify possible miscalculation inaccuracies, and therefore, estimates current consumption patterns by using multiple databases.

The measurement and monitoring of food environments (variety of foods available, affordability, convenience and desirability for the population) through the evaluation of diet quality play a key role in establishing effective policies, as in the case of sugary drinks for example [21]. Reliable data on food consumption are, therefore, necessary to better explain the impacts of dietary changes and the progressive shift away from the Mediterranean diet on the health of Spanish people. Indeed, as noted by other authors, it is essential to assess food consumption and diet quality, as such an appraisal allows the identification of the dietary patterns required to improve the nutritional status of Spain’s population [22,23].

Lastly, a review was performed by del Pozo de la Calle et al. (2015) of the different tools available to collect information on community feeding at the national, household or individual level. The authors highlighted the limitations and strengths of the different methods but did not support their conclusions with quantitative data, so their findings remain theoretical [24]. Therefore, it seems that the best approach to carry out this type of analysis is that the population consumption pattern remains unclear.

The objective of the present study was twofold. Firstly, we sought to quantify inaccuracies in the official data by developing a method to quantify the food consumption of the Spanish population using secondary data from official sources, overcoming the limitations and inconsistencies observed in other studies mentioned above. Secondly, we compared the population’s current food consumption patterns based on our approach with today’s dietary recommendations by conducting a nutritional analysis

## 2. Materials and Methods

**Design and sources.** Secondary data obtained from public sources were analysed. This type of design is exempt from the requirement of approval by an ethics committee.

Figure 1 illustrates the methodological design used to calculate the divergence between Spanish food consumption according to official public sources dedicated towards reporting this information, and our own estimation using new public information intended for other purposes, e.g., usage by commerce or industry.

All the information used was obtained from different public access databases. The bases used in each phase of the analysis, as well as the possible paths that any raw material can take to the point of sale are shown in Figure 2:

**Data analysis procedure.** The methodology employed required a multitude of calculations to determine balances of food produced, marketed, consumed, or used for purposes other than human consumption. For this reason, in this section, we present the calculations that we believe are essential to understand the study. We refer readers to the Appendix A for further information on tourist food consumption, agrarian production, disaggregated Spanish industrial production data, raw materials and processed products conversion, foreign trade, agricultural and industrial production correspondence, and food chain reconstructions (See Appendix A supported by references [25,26,27,28,29,30,31,32,33,34,35,36,37,38,39,40,41,42,43,44,45,46,47,48,49,50,51,52,53]).

The main calculations performed on this basis are detailed below:

**(a) Calculation of the estimated official food consumption (EOFC).** First, we calculated the official gross food consumption (Equation (1)) using the data of the Food Consumption Panel of the Ministry of Agriculture, Fisheries and Food (2017), completed with the extra-domestic consumption data from the 2017 Report on food consumption in Spain (Ministry of Agriculture, Fisheries and Food, 2017) [25]. In addition, the balance of tourist food consumption was also considered (consumption of the tourists who visit the country vs. Spanish tourists who are absent from the country, see Appendix A). Finally, food losses and waste during food distribution were taken into account according to a previously described methodology [37]. These data were calculated for the 34 most representative products, which, according to the Panel, accounted for 76% of the volume consumed. The remaining 24% showed great similarities with the domestic/extra-domestic consumption relation of the considered foods (e.g., the extra-domestic consumption of garlic was almost identical to that of onion) [37]. In other cases, we assumed, on the contrary, that the official consumption (Equation (1)) was equivalent to the final apparent consumption (e.g., honey).
(1)$$EOFC_{}=\frac{C_{panel}}{1-FLW_{distrib}}+C_{panel}.{\%}_{extradom}.{\left(1-{\%}_{extradom}\right)}^{-1}.{\left(1-FLW_{distrib}\right)}^{-1}$$

Equation (1), where EOFC: estimated official food consumption, Cpanel: household consumption indicated by the Food Consumption Panel, FLWdistrib: losses in food distribution and %extradom: estimated extra-domestic consumption factor [37]

Next, the final apparent consumption (FAC) was calculated for the 34 products, defined as the amount of food obtained from the reconstruction of the food chains destined for the Spanish population (Equation (2)). The FAC was obtained considering that: (a) A share of the products is consumed fresh and another part is used as industrial raw material; (b) Industrial processing (i.e., dehydration by curing, cooking, concentration, dilution, etc.) modifies the volume of food products available for human consumption; (c) Some products (i.e., animal fats for feed production) are not intended for human consumption
(2)$$FAC=P+I-E-\overset{i}{\underset{0}{\sum }}PCF_{n}-\overset{i}{\underset{0}{\sum }}FLW_{n}$$

Equation (2), where P: national production, I: import, E: export, PCF: processed and conserved food, and FLW: losses and waste throughout the chain for each phase of the food chain considered (Hui, 2012 [54]).

The necessary information for the rebuilding of food chains and the different uses of their products was obtained through expert knowledge, academic literature, *Codex Alimentarius* protocols, consultations with agents of the sector and using food industry manuals and protocols [34,40,54,55,56].

The Agrarian Statistics Yearbook [57] and the Livestock Statistics [58] and Industrial Production Survey [27] (for semi-transformed and transformed products) were used to calculate the national production. Import and export data were obtained from DATACOMEX (Ministry of Industry, Trade and Tourism, 2017) [39]. Losses and waste in the food chain according to FAO (2011) [37] were also taken into account.

The difference observed between FAC and EOFC for each product allowed us to calculate the percentage of increase (Equation (3)) used as a corrective factor of data published by the MAPA, thus establishing the estimated real food consumption (ERFC).
(3)$$\%Increase=\frac{\left(FAC-EOFC\right)}{EOFC}$$

Equation (3), where %Increase: percentage of increase, FAC: final apparent consumption, and EOFC: estimated official food consumption

(b) Analysis of adherence of ERFC to a healthy eating pattern. Consumption patterns were compared with the current national recommendations [59] by calculating the ratio between consumed servings and recommended servings (Rcr). Due to sociodemographic variability, we considered that an Rcr interval of 0.8–1.2 would allow the maintenance of a good state of health of the population.

Additionally, a nutritional analysis using the DIAL v3 (Alce Engineering) software was performed to assess whether ERFC satisfied the needs of a man and woman aged 42 years—because this age group corresponds to the largest segment of the population (38–51 years) according to the National Statistics Institute [60]—as considered by the Nutritional Assessment of the Spanish Diet report. The rest of the considerations were a weight of 68 kg and a height of 1.70 m (BMI = 23.5 kg·m^−2^) for the man, and 65 kg and 1.65 m (BMI = 24 kg·m^−2^) for the woman. A healthy BMI was considered to avoid peculiarities in food and nutritional recommendations associated with overweight/obesity. We considered a light physical activity factor of 1.374 (that is, light exercises 1–3 times/week) due to the fact that 46.5% of men and 54.8% of women do not dedicate any day of the week to physical exercise during their leisure time and that 37.8% of the Spanish population aged over 15 years declares itself sedentary [61]

Energy requirements were obtained by multiplying the basal metabolic rate (Harris Benedict’s Equation (4)) by the light activity factor [62].
BMR (men) = 66.5 + (13.76 × kg) + (5.003 × cm) − (6.755 × years)BMR (women) = 655 + (9.563 × kg) + (1.850 × cm) − (4.676 × years)
(4)

The consumption of macro- and micro-nutrients was compared with the current recommended daily intake (European Food Safety Authority, 2017) [63] and latest consensus recommendations on fats and oils in the diet of the adult Spanish population [64].

## 3. Results

Regarding the EOFC, Table 1 presents the values obtained according to Equation (1) and the percentage increase according to Equation (3) for each food group. All groups achieved a positive increase except the Fish and Nuts Groups.

Table 1 also shows the extent to which the estimated diet followed a healthy eating pattern, as well as the ERFC, the consumption recommendations, and the ratio between the two. We considered an acceptable range of consumption (0.8–1.2) and observed that cereals, dairy products, fish and eggs were within the range, while meat was three times above the upper limit. The other food groups were all below the range, Nuts accounting for the lowest ratio, followed by Legumes and Fruits, whose consumption represented half the recommended amount.

We also analysed the consumption of foods that were not the subject of any specific consumption recommendations, as well as foods recommended to be consumed occasionally. The results are shown in Table 2.

Notable among the occasional consumption foods were prepared dishes, buns and, in particular, fermented alcoholic beverages and soft drinks.

The results of the nutritional analysis regarding macro- and micro-nutrients are shown in Table 3 and Table 4, respectively. We calculated the corresponding percentages of their contribution to the nutritional needs of men and women.

The kilocalories provided by the diet exceeded the needs of men and women by 36% and 54%, respectively. Protein consumption was 41% and 65% higher than that recommended by the WHO (0.8–1 g·kg^−1^·day^−1^) in men and women, respectively. Despite this, like the rest of the macro-nutrients, the percentage of kilocalories provided by protein (14.6%) fell within the EFSA recommendations. However, the lipid profile reflected an excessive consumption of saturated fat and cholesterol, while mono- and poly-unsaturated fats remained within recommended amounts. Fibre intake was adequate, although carbohydrate intake was slightly lower than recommended.

Table 4 illustrates the micro-nutrient intake. Deficits in iodine and vitamin D can be observed, as well as an excess of phosphorus.

## 4. Discussion

In this study, we described a new exhaustive methodology to calculate the food consumption of the Spanish population. This technique allowed the correction of the values issued by official sources and explicated the gap between them and the apparent consumption data offered by sources such as FAOSTAT or other researchers [17,65]. In addition, we quantified the degree of compliance with healthy diet recommendations and provided a nutritional analysis of a typical man and woman with a healthy weight. Together, this information has an important value for public health as it can be used to design programmes aimed at improving community nutritional health.

Eating habits are influenced by fashions, publicity, and the availability of new exotic foods. The evolution of food consumption patterns in recent decades has come with higher mortality and morbidity rates in developed countries, creating a serious public health problem [66]. In this sense, it is necessary to have reliable data on the nutrition and food patterns in each country. We must be aware, nevertheless, of the difficulties in building accurate knowledge of dietary intakes. For more than 20 years the Food Consumption Panel has provided questionable data as it has been relying on calculations of food consumption in homes, hotels/restaurants and institutions using data recorded via the scanning of purchased products. The methodology implemented in this work, however, shows that official sources do not actually capture all food consumption.

Our results revealed that generally, food consumption is higher than the figures reported by the official Consumption Panel. In addition, the recommendations of some food groups fail to be met even despite such higher intakes. The amount of fruit and vegetables was greater than that reported by the Panel, yet the current recommendations of five servings a day (three fruits and two vegetables) are still not being fulfilled, according to what has been previously reported (i.e., the amounts slightly exceed two servings) in the latest national evaluation of the consumption of these foods [67]. Similarly, the consumption of fruit and vegetables was 20% higher than that reported by the Panel, but only half the recommended weekly portions of legumes were consumed and much less than that (Rcr = 0.14) in the case of nuts.

The greatest difference was observed in the case of meat: consumption was 50% greater than the figure published by the panel. This high meat consumption rate together with the observed fibre intake insufficiency could explain Spain’s large number of cases of colorectal cancer [68]. The analysis altogether reflects how Spanish eating patterns have today moved far from the traditional Mediterranean diet—which is mainly plant-based and rich in fibre, micro-nutrients, as well as bioactive compounds with antioxidant activity. Such a situation could, in fact, be related to Spain’s high rates of cardiometabolic problems (obesity, diabetes, and cardiovascular diseases) [69].

Historically, carbohydrate intakes in the Spanish diet came mainly from grain and derivatives. Nowadays, the lower amounts observed in the diet of cereals and derivatives may be due to the consumption of white bread, which might increase body weight, and/or because the bread consumed is of a lower quality [70]. Bread consumption was 368 g/day in 1964 vs. 139 g/day in 2012, reflecting a gradual decline over time. Our results indicated a Rcr = 1 for the group of cereals, and we found a 51% increase in consumption with respect to official data. This consumption rate can be regarded as reasonable given that we took a population that is not very active as a reference, and that currently the consumption recommendations for this food group in Spain are linked to physical activity [59]. The diminished consumption of cereals and the low consumption rates of vegetables and legumes, however, that are also a source of healthy carbohydrates, could suggest that the intake of these macro-nutrients is considerably below the desirable amount (42% vs. 45–60%). Other national and recent studies about the Spanish diet have revealed similar percentage of kilocalories provided by carbohydrates (41% [13,18] and 42% [14]). Regarding to fibre intake, our results (27.7 g/day) have been higher than previous studies that revealed amounts of 19 g/day [13], 18 g/day [14], and 25 g/day [71]. Carbohydrates could have been displaced by the consumption of lipids that have resulted in 40%, the highest recommended limit (20–40%). Similar results have been obtained in previous studies: the percentage of kilocalories provided by Lipids was 41%, 43% and 38.5%, respectively [13,14,18]. These results could be explained because the observed daily consumption (rather than the recommended sporadic and moderate consumption) of energy-dense and atherogenic ultra-processed foods, whose nutritional contributions are poor (high energy density, low fibre and micro-nutrient content and high amounts of saturated and trans fats, and simple sugars) [71]. Specifically, we observed high consumption rates of soft drinks, prepared dishes and pastries, along with wine and beer (Table 4). This result is in line with the findings of Monteiro et al. (2018) who analysed the distribution (%) of total household food availability (kcal·person^−1^·day^−1^). They found that 26.4% kcal/day came from the consumption of ultra-processed foods [72]. In 2010, ultra-processed foods represented 31.7% of daily energy acquisitions according to recent Spanish data [73]. Currently, this class of foods represents the largest source of energy intake in some countries, and this pattern of consumption is linked to an increase in obesity rates, favouring the existence of obesogenic environments [74]. However, data published in Spain on dietary caloric intake point to a decrease over the last few years. Average energy intake (domestic and extra-domestic) was 2761 kcal/day in 2006, according to an assessment based on the Food Consumption Panel, while the estimated caloric intake was 2609 and 2358 kilocalories a day, respectively, in 2012 and 2019 [13,14,75]. Our estimation corresponded to an intake of 2930 kcal/day, which we deem more accurate, considering the high intake of ultra-processed foods, characterised by large amounts of free sugars and saturated fats, which all contributes to increasing energy intake [76]. In addition, this excessive caloric intake—which goes well beyond the population’s needs –would partly explain Spain’s high prevalence of overweight and obesity in children, adolescents, and adults [3,4].

Beyond the caloric intake, our results showed that the percentage of saturated fatty acids surpassed the recommendations (12.5% of the caloric intake). Similar results to these have been published for Spain [18] revealing a worsening of the macronutrient distribution. Regarding cholesterol, the international recommendations are to consume as little dietary cholesterol as possible. We found an excessive intake (432 mg/day) comparable with previous data that revealed a daily consumption of 454.8 mg in 1876 men from Aragon [71]. Excessive intakes of cholesterol together with the overconsumption of saturated fats, could be contributing to the high prevalence of hypercholesterolemia [9] and diet-related non-communicable diseases [2].

Finally, with respect to micro-nutrients, our results showed a deficit in the intake of iodine and vitamin D (117 mg/day and 3.4 µg/day, respectively). While vitamin D intakes are frequently under needs (5.3 µg/day [71]), this micronutrient could be covered by sun exposure. However, iodine deficiency could be a major public health problem due to excess mortality associated with moderate/severe iodine deficiencies recently highlighted in Spain’s adult population [77]. Additionally, the phosphorus intake excess could have adverse health effects [78]. The high intake of phosphorus could be associated with the greater consumption of highly processed foods, especially fast foods and convenience foods. This is a fact that we should not lose sight of, as it could have implications for bone health and, as recently suggested, for cardiometabolic health through increased peripheral insulin resistance and altered levels of weight-regulating hormones [79].

**Reliability and validity.** The Industrial Products Survey and Spanish agricultural statistics maintain a strong reliability derived from: (a) Cross-synergies with other statistical operations, such as the Common Agricultural Policy mandatory field notebooks, the National Structural Survey, the Surface and Yield Agricultural Survey, the GIS approaches (SIGPAC), among others; (b) The administrative level on which they relies to extract data, which has as a source of information administrative registers, direct surveys, and georeferencial samples.

On the other hand, Commercial Exchange Statistics (DATACOMEX) related with import/export fluxes rely on economic performances of individual businesses that must declare goods and services supplied to and from third countries. In this sense, the accuracy of this database seems accurate enough.

As for coefficients used to calculate food waste and losses, FAO (2011) is the most common reference used by most of the assessments related with environmental performances and up to date there is has noy been any study that could increase the level of accuracy. Nevertheless, errors derived from wrong FLW estimations outrange the level of inconsistency between final apparent consumption and official calculated consumption.

External validity refers to the extent and way in which the results of an experiment can be generalized to different subjects, populations, places, experimenters, etc. In this case, as in most studies where there are cultural, organizational or political components, external validity is very difficult to guarantee. Different countries have different protocols for collecting information, for its publication and for public use; therefore, we encourage other authors to explore this method, but we do not guarantee that it will be reproducible since we do not know the organizational aspects of each country.

**Limitations**. Despite the importance of the results presented, several limitations to this study must be highlighted, some limitations and source of errors during the analysis were identified, especially in relation with equivalences between commerce codes (TARIC) and agricultural/livestock products, due to the absence of a document that clarifies the equivalence as happens with industrial production. Nevertheless, we corroborated the equivalence by analysing 2017 ± 5 years to corroborate tendencies and equivalences (not included in the results).

We also identified some errors in the MAPA report from which extra-domestic consumption data was obtained, as they indicate different values for the same food categories within the same document. When this happened, we chose the most suitable value based on specific known patterns of consumption for the product. As for the FAO FLW coefficients, we acknowledged the error derived from using a coefficient for a whole continent, but up to date it is the best-known approach for measuring food losses and waste. Even more, results provided here describe a bigger magnitude of difference between official consumption and final apparent consumption volumes, therefore, reducing the level of inaccuracy because of a non-suitable FLW coefficient.

Finally, there is a limitation to consider in the nutritional analysis. DIAL software does not calculate free/added sugars and it could be interesting to complete our results, although we do not believe these data are indispensable.

**Policy implications**. The FAO identified some time ago, in 1992, that to improve nutrition internationally, it was essential to assess, analyse, and monitor nutrition situations, as well as to incorporate nutrition objectives into development policies and programmes (FAO/WHO, 1992) [80]. Since then, there has been a growing interest in analysing the food consumption of a given population. Indeed, there are further motives to evaluate this information beyond the traditional objective of assessing the nutritional quality of a diet. In both Spain and other countries, this information also helps to determine the degree of sustainability of a country’s diet and the environmental impact of the food system that supports it [81]. Because policy makers may use this information to establish sanitary, commercial or environmental regulations, it should be as reliable as possible [82]. Food policy formulations must be guided by a periodic evaluation and re-evaluation of food systems. The latter will lead to greater accuracy, taking into account the multidisciplinary work required to conduct such an evaluation [83]. A clear example is the EU’s Farm to Fork Strategy, which aims at making food systems fair, healthy, and environmentally friendly, promoting diet changes, among other measures [84]. Although this strategy is common to all member states, each one must develop, implement, and subsequently, evaluate the initiatives that it considers most appropriate based on its starting situation. In the case of Spain, the work presented here contributes to this, as it provides more realistic information on the average Spanish diet today and a more precise evaluation methodology to capture the changes that will occur in the future.

## 5. Conclusions

We believe that the estimates provided in this study offer a more realistic view of Spanish food consumption. They are also more consistent with the country’s data on nutritional and health problems as described both by government agencies and the literature. The results of the study indicate that the Spanish Consumption Panel may be underestimating the population’s diet. The present work also provides relevant alternative information that reveals how consumption is generally higher than reported. Despite this, consumption recommendations are not met by some food groups, while in other groups consumption is much higher than recommended amounts.

It is necessary to use these data when developing effective food guides, as well as when designing new training, communication, and dissemination strategies on food and nutrition. According to our results, ideally, a higher consumption of fruits, vegetables, legumes, and nuts should be recommended to the detriment of highly industrially processed products in general, and of meat origin in particular. This way, an important step would be made towards disseminating the Mediterranean diet pattern whose benefits both for public health and the environment have been widely proven.

We encourage other countries to explore their panel data in order to assess possible biases, as well as to develop common methodologies that allow decision-making in matters of governance, both nationally and at the level of the European Union.

## Figures and Tables

**Figure 1 ijerph-20-01642-f001:**
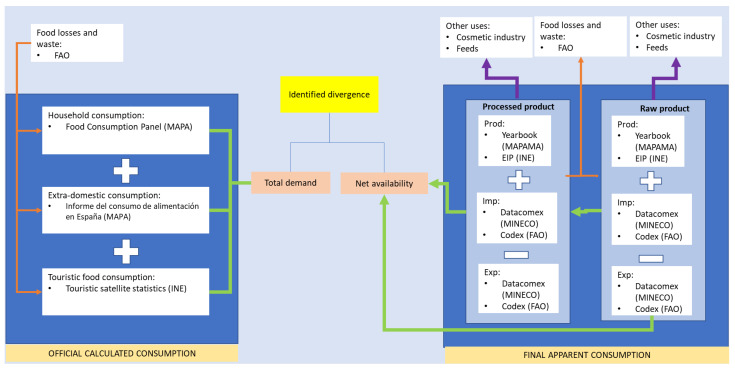
Methodological approach to calculate official consumption and final apparent consumption. The orange lines are the computed loss streams (added or subtracted). The green lines represent the amount of product that passes to the next phase of the food chain. The purple line indicates the output of the product to other uses. The existence of at least two generic phases in the transformation process was considered. They are related to the production of processed foods since they use part of the agricultural and livestock products for their production processes, although these phases may be more numerous. An example would be processed tomato, such as fried tomato, which has at least two consecutive phases: transformed tomato paste is used in its production (whose original raw material was fresh tomato in a previous phase), and crushed fresh tomato, combined in variable proportions (see Appendix A supported by references [25,26,27,28,29,30,31,32,33,34,35,36,37,38,39,40,41,42,43,44,45,46,47,48,49,50,51,52,53]).

**Figure 2 ijerph-20-01642-f002:**
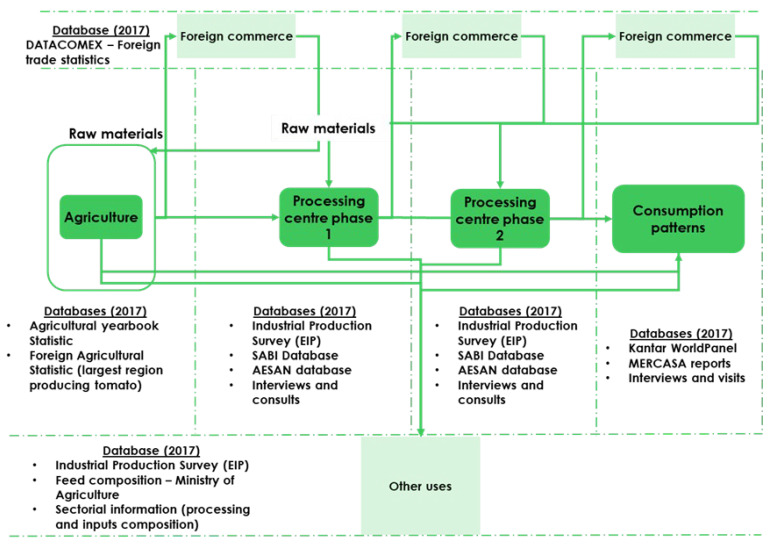
Diagram of the databases and paths to the point of sale. DATACOMEX: Foreign trade statistics of goods from Spain and the EU, SABI: Spanish acronym for Iberian Balance Sheet Analysis System, it provides information about Spanish companies or groups of companies, AESAN: Spanish acronym for Spanish Agency for Food Safety and Nutrition, MERCASA: a public company of the State Administration that has been providing a public service to the entire food chain since 1966, structuring the largest network of wholesale markets in the world, which covers the whole of Spain’s territory. (See Appendix A supported by references [25,26,27,28,29,30,31,32,33,34,35,36,37,38,39,40,41,42,43,44,45,46,47,48,49,50,51,52,53]).

**Table 1 ijerph-20-01642-t001:** Values obtained from the components of Equation (3) and ratio of consumed/recommended servings.

Food (g/Serving)	EOFC (gd^−1^p^−1^)	%Increase	ERFC (gd^−1^p^−1^) [Serving]	Rec Cons [Serving]	Rcr
Vegetable (200 g)	233.2	23%	285 [1.4/day]	2/day	0.7
Fruits (200 g)	267.3	12%	300 [1.5/day]	3/day	0.5
Cereals *	232	51%	350 [4.5/day]	4/day **	1.0
Milk (dairy) *	212 (109)	25%	215 (186) [2.5/day]	2–4/day	0.8
Oils (10 g)	32.7	5%	34 [3.4/day]	4–6/day	0.7
Meat (125 g)	131.7	50%	200 [11/week]	3/week	3.6
Fish (125 g)	74	0%	74 [4.0/week]	3–4/week	1.0
Legumes (70 g)	12.3	17%	14.3 [1.4/week]	2–4/week	0.5
Eggs (60 g)	27.1	39%	34.3 [4.0/week]	3–5/week	1.0
Nuts (25 g)	2.56	0%	2.56 [0.7/week]	3–7/week	0.14

* Due to the different weight/serving of each type of food, the number of consumed servings were calculated separately (according to Appendix A in the Appendix A), considering the sum of the group to determine the Rcr.** For a physical activity factor of 1.375, the recommendation is 4 servings/day, %Increase: percentage of increase, EOFC: estimated official food consumption, (gd^−1^ pc^−1^): grams·day^−1^·person^−1^, ERFC: estimated real food consumption, Rec Cons: recommended consumption, Rcr: ratio of consumed/recommended servings.

**Table 2 ijerph-20-01642-t002:** Consumption of foods recommended to be consumed occasionally.

Food	ERFC (gd^−1^ p^−1^)
Sugar and honey	12
Chocolates	12.5
Buns	45
Butter	3
Sauces	11
Prepared dishes	58
Snacks	6
Soft drinks	166
Wine and beer	285
Other alcoholics drinks	7

ERFC: estimated real food consumption, (gd^−1^ pc^−1^): grams·day^−1^·person^−1^.

**Table 3 ijerph-20-01642-t003:** Macro-nutrients’ analysis according to estimated real food consumption.

Nutrients	Intake	RDI		%Contribution	
		Man	Women	Man	Women
Energy (kcal/day^−1^)	2930	2156	1906	136%	154%
Proteins (g/day^−1^)	107	61–76	52–65	141%	165%
**Caloric profile**		**Recommended ***			
Proteins (%kcal)	14.6%	10–15%			
Lipids (%kcal)	40%	20–40%			
Carbohydrates (%kcal)	42%	45–60%			
**Lipid profile**		**Recommended ****			
SFA (%)	12.5%	7–8%			
MUFA (%)	17.3%	13–18%			
PUFA (%)	6.7%	5–10%			
Cholesterol (mg/day^−1^)	432	300			
Fibre (g/day^−1^)	27.7	25–30			

* According to the European Food Safety Authority, EFSA, ** according to the FESNAD (Spanish acronym for Spanish Federation of Nutrition, Food, and Dietetics Societies), RDI: recommended daily intake, SFA: saturated fatty acid, MUFA: monounsaturated fatty acid, PUFA: polyunsaturated fatty acid

**Table 4 ijerph-20-01642-t004:** Micro-nutrients’ analysis according to estimated real food consumption.

Micronutrient	Intake	RDI	%Contribution
Calcium (mg)	955	950	100%
Phosphorous (mg)	1356	550	246%
Magnesium (mg)	351	M: 350	100%
		W: 300	117%
Zinc (mg)	10.1	M: 9.4	107%
		W: 7.5	135%
Iron (mg)	21.2	M:11	193%
		W: 16	132%
Iodine (mg)	117	150	78%
Vitamin B1 (mg)	1.6	0.75	213%
Vitamin B2 (mg)	1.9	1.6	119%
Vitamin B6 (mg)	2.8	M: 1.7	165%
		W: 1.6	175%
Vitamin B9 (mg)	324	330	98.2%
Vitamin B12 (µg)	12.2	4	305%
Vitamin C (mg)	154	M: 110	140%
		W: 95	162%
Vitamin A (µg)	1041	M: 750	139%
		W: 650	160%
Vitamin D (µg)	3.4	15	23%
Vitamin E (µg)	18.5	M: 13	142%
		W: 11	168%
Vitamin K (µg)	276	70	394%

µg: micrograms, mg: milligrams, RDI: recommended daily intake, M: man, W: woman.

## Data Availability

Not applicable.

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
