# Peer review of "Nutritional Analysis of the Spanish Population: A New Approach Using Public Data on Consumption"

_ijerph, 2023, doi:10.3390/ijerph20021642_

Round 1

Reviewer 1 Report

The manuscripts entield: "Nutritional analysis of the Spanish population: a new approach using public data on consumption" provides an interesting approach regarding new tools to evaluate the underestimation of offical food consumption data, baed on the cross with data from food losses and wastes along the distribution chain. Moreover, this study also analyses the nutritional adequacy of the Spanish population diet based on the existent international and national dietary guidelines.

The objectives of the developed work are clearly and were accomplished. Concerning the methodology applied, it is recommended to summarize the information, in order to other countries are able to perform it. Something like, a flowchart probably will be useful.

The results and dicussion are appropriate. Therefore, in my opinion the manuscript deserves to be published in International Journal of Environmental Research and Public Health after minor revision.  

Author Response

Before beginning to respond to the comments, the authors wish to thank the reviewer for the effort and time invested in reading the manuscript. We also appreciate the words of approval you have shown. Regarding your proposal to add a flowchart as a summary of the procedure carried out, it seems correct to us and it has been included in a new version of the supplementary material.

Reviewer 2 Report

1Dear Authors,

It was a pleasure to review your manuscript. The manuscript can be considered for publication, providing amendment/clarification of the following points:

Abstract should answer to three conceptual questions as what was done, what was found and what was concluded? This is currently not the case as the aim is unclear and instead there an attempt to establish the rationale, but it is not easy to visualise what was actually done, how it was done, what are the results found and what is the conclusion.

2.       Line 58 – 60: Why? Please elaborate?

3.       There is an attempt to establish a gap in the knowledge and a rationale for this study, but that needs major improvement.

4.       The introduction does barely include a literature review. If the results are showing an overconsumption of meat and derivatives, soft drinks, prepared dishes and pastries, along with wine and beer, and an underconsumption of fruits, legumes and nuts, together with caloric intake surpassing needs, what is the current body of knowledge on each regardless of the current study. To address this point, you might want to produce a table with studies, authors and date, outline methods, findings, strengths and limitations to support the literature review.

5.       Lines 73-76, what does it mean? What were the findings? What does it add here?

6.       Lines 77-82, there is a major difference between the aims and objectives. Please review and clarify.

7.       Line 84: Secondary data obtained from public sources were analysed. Question A: What data and what public sources? Question B: Why should we believe that this provides the most complete picture of the nutritional status for the Spanish population? Question C: Comment on the reliability and validity of data used. Question D: Please clarify how you made sure about the methodological reliability and validity (internal and external validity) of the data.

8.       Line 85: This type of design is exempt from the requirement of approval by an ethics committee. Question A: Please provide official letter from the ethics committee that the research does not warrant ethical approval. Question B: Please provide a clear statement to confirm that the original research ethics permitted such additional analysis and you have been authorised to use the data. Question C: Please elaborate how the ethical approval was achieved for all primary data collection.

9.       Lines 84-9: The narrative need clarification and improvement. The paragraph aims to provide information on design and sources getting assistance from Figure 1 on Methodological approach to calculate official consumption and final apparent consumption. Question A: It is not clear why this approach was needed to be taken? Question B: Why this is rigorous (reliability and validity)? Question C: Has this been used elsewhere? Question D: How the conceptual framework was developed and validated?

10.   Lines 121-180: Reading about the equations, the reader wonders if there are reliable national Diet and Nutritional Surveys in the Spain assessing the dietary intake (comparable to NHANES, NDNS, German Nutrition Survey)? If so, why there is a need to current approach?

11.   Lines 174-176: ‘The rest of the considerations were a weight of 68 kg and a height of 1.70 m (BMI = 23.5 kg*m-2 for the man and of 65 kg and 1.65 m (BMI=24 kg*m-2 for the woman. A healthy BMI was considered to avoid peculiarities in food and nutritional recommendations associated to overweight/obesity.’ Please clarify how this is comparable to Spanish/EFSA reference man/woman used for estimated average energy and nutritional requirements?

12.   Major overall question: The study claims that it describes a new, exhaustive methodology to calculate the food consumption of the Spanish population, however, altogether it confirms that findings of the previous studies (i.e. there is an overconsumption of meat and derivatives, soft drinks, prepared dishes and pastries, along with wine and beer, and an underconsumption of fruits, legumes and nuts. Caloric intake surpasses needs, entailing an excess of non-communicable diseases), which is the case for most developed countries. What are the new findings?

13.   Tables 3 and 4 demonstrate that overall the intake meet RDI of almost all (apart from Iodine and Vitamin D, for obvious reason) micronutrients and almost all (with exception of the fats and lipids) for macronutrients. Question A: Why free/added sugars are not reported? Question B: Does it mean that with excessive energy intake, the overall nutritional requirement has been offset? Question C: What is the translation of these findings and how do they relate with nutritional toxicities, deficiencies and disorders previously reported  within/from the Spanish population?

14.   Please comment on supplement consumption, how considered within the current analysis and what is the estimated/potential contribution to overall macro and micronutrient intake?

15.   As per introduction: ‘The objective of the present study was twofold. First, we sought to develop a method to quantify the food consumption of the Spanish population using secondary data from official sources, overcoming the limitations and inconsistencies observed in other studies mentioned above. The second aim was to compare the population’s current food consumption patterns with today’s dietary recommendations by conducting a nutritional analysis’. Reading the article, with regard to objective 1; methodologically, I am not fully convinced why new methods should be developed the way that is developed? Why this is valid and reliable? With regard to aim 2; I am not fully clear how does it compare with the findings of the previous studies? And in comparison with other countries? For instance, the fiber intake calculated by this method is encouraging but does it correspond with rest of literature and our understanding of carbohydrate intake and its consequences within the Spanish population?

16.   There are no discussions on limitations and sources of errors within the current work, please clarify. Please also explore the strategies used/that can be used for enhancing the rigour.

Author Response

Before beginning to respond to the comments, the authors wish to thank the reviewer for the effort and time invested in reading the manuscript. 

Reviewer 3 Report

This article present nutritional analysis of the spanish population using public data on consumption. It will help the scientific community to collect reliable data on food consumption and analyzing associated factors in the specific community.  Before recommending this article for publication, there are some shortcomings for that should be resolve.

This section is well written, but the author has focused too much on the results part. The authors should present summarized methods in the abstract section.

In addition, the sentences are very long which void the main concept of the sentences.

All results are generally described in the abstract. Specific or quantitative results should be presented in the abstract  

In the last section add one to two sentences of conclusion and future recommendation.

Line 37 should be cited with recent and relevant studies. The following studies would be helpful.

https://doi.org/10.3390/molecules27196728,

Add subsections in the methods section

From which departments data was collected and what was the process should be mention in the main file.

Add “Harris Benedict’s equation” in line 182-183.

Discussion is well justified however the authors should present and discuss some relevant studies with their results.

Author Response

Before beginning to respond to the comments, the authors wish to thank the reviewer for the effort and time invested in reading the manuscript. We also appreciate the words of approval you have shown. 

Round 2

Reviewer 2 Report

The manuscript has improved enough to be of publishable quality, although there are still key limitations that could have been improved to strengthen the manuscript.